# Concordance between SIVA, IVAN, and VAMPIRE Software Tools for Semi-Automated Analysis of Retinal Vessel Caliber

**DOI:** 10.3390/diagnostics12061317

**Published:** 2022-05-25

**Authors:** Thibaud Mautuit, Pierre Cunnac, Carol Y. Cheung, Tien Y. Wong, Stephen Hogg, Emanuele Trucco, Vincent Daien, Thomas J. MacGillivray, José Labarère, Christophe Chiquet

**Affiliations:** 1HP2 Laboratory, INSERM U1300, Univ. Grenoble Alpes, 38700 La Tronche, France; t.mautuit21@orange.fr (T.M.); pierre.cunnac@gmail.com (P.C.); 2Department of Ophthalmology, University Hospital of Grenoble Alps, CEDEX 09, 38043 Grenoble, France; 3Department of Ophthalmology and Visual Sciences, The Chinese University of Hong Kong, Hong Kong, China; carolcheung@cuhk.edu.hk; 4Singapore Eye Research Institute, Yong Loo Ling School of Medicine, National University of Singapore, Singapore 119077, Singapore; ophwty@nus.edu.sg; 5VAMPIRE Project, School of Computing, University of Dundee, Dundee DD1 4HN, UK; s.c.z.hogg@dundee.ac.uk (S.H.); e.trucco@dundee.ac.uk (E.T.); 6Department of Ophthalmology, Gui De Chauliac Hospital, 34295 Montpellier, France; v-daien@chu-montpellier.fr; 7VAMPIRE Project, Clinical Research Imaging Centre, University of Edinburgh, Edinburgh EH8 9YL, UK; t.j.macgillivray@ed.ac.uk; 8Clinical Epidemiology Unit, Grenoble University Hospital, 38043 Grenoble, France; jlabarere@chu-grenoble.fr; 9TIMC-IMAG UMR 5525, CNRS, Univ. Grenoble Alpes, 38041 Grenoble, France

**Keywords:** central retinal artery equivalent, central retinal vein equivalent, SIVA software, IVAN software, VAMPIRE software, conversion algorithm, retinal vessel measurements

## Abstract

We aimed to compare measurements from three of the most widely used software packages in the literature and to generate conversion algorithms for measurement of the central retinal artery equivalent (CRAE) and central retinal vein equivalent (CRVE) between SIVA and IVAN and between SIVA and VAMPIRE. We analyzed 223 retinal photographs from 133 human participants using both SIVA, VAMPIRE and IVAN independently for computing CRAE and CRVE. Agreement between measurements was assessed using Bland–Altman plots and intra-class correlation coefficients. A conversion algorithm between measurements was carried out using linear regression, and validated using bootstrapping and root-mean-square error. The agreement between VAMPIRE and IVAN was poor to moderate: The mean difference was 20.2 µm (95% limits of agreement, LOA, −12.2–52.6 µm) for CRAE and 21.0 µm (95% LOA, −17.5–59.5 µm) for CRVE. The agreement between VAMPIRE and SIVA was also poor to moderate: the mean difference was 36.6 µm (95% LOA, −12.8–60.4 µm) for CRAE, and 40.3 µm (95% LOA, 5.6–75.0 µm) for CRVE. The agreement between IVAN and SIVA was good to excellent: the mean difference was 16.4 µm (95% LOA, −4.25–37.0 µm) for CRAE, and 19.3 µm (95% LOA, 0.09–38.6 µm) for CRVE. We propose an algorithm converting IVAN and VAMPIRE measurements into SIVA-estimated measurements, which could be used to homogenize sets of vessel measurements obtained with different software packages.

## 1. Introduction

Since the retinal vasculature reflects systemic microvascular damage, the analysis of retinal fundus photographs offers a noninvasive method for estimating cardiovascular risk. Several health outcomes are associated with changes in the diameter of retinal vessels, such as systemic hypertension, stroke, and non-pathological cognitive aging [1,2,3,4,5,6,7,8,9].

A quantitative description of the retinal vessel network can be provided by several semi-automatic software tools. Three common software packages reported in the literature are IVAN (Integrative Vessel Analysis), [10,11,12,13,14] SIVA (Singapore I Vessel Assessment), [15,16] and VAMPIRE (Vascular Assessment and Measurement Platform for Images of the Retina) [17,18,19]. All of them include three well-known parameters summarizing vessel widths around the optic disc: CRAE (central retinal artery equivalent), CRVE (central retinal vein equivalent), and arteriovenous ratio (AVR). However, the software packages use different algorithms for vessel segmentation, for vessel labeling (artery or vein), and for estimating vascular diameter. Two additional features are estimated with SIVA and VAMPIRE but not with IVAN: tortuosity and fractal dimension.

Previous studies suggest poor-to-moderate concordance between these software packages. A comparison between SIVA and VAMPIRE on 655 images indicated poor-to-limited agreement for all parameters (0.16–0.41) and the presence of proportional and systematic bias in the majority of the parameters [20]. IVAN, SIVA, and RA (Retinal Analysis) have been compared on a series of 120 retinal photographs, but the intra-class correlation to assess concordance was not reported [21]. Nevertheless, the authors showed that IVAN yields significantly larger retinal vessel measurements than SIVA and they proposed an algorithm to convert IVAN measurements into estimated SIVA measurements. This approach seems attractive for pooling data from different studies.

The poor concordance that has been reported implies that the software packages are not interchangeable. There is a need to clarify the quantitative differences between their measurements, which may lead to different conclusions in statistical studies using the same image set. This calls for a systematic and structured comparison of these three software applications.

The primary aim of our study was to determine the agreement of CRAE, CRVE, and AVR estimates from the three software tools. Our secondary aim was to propose an algorithm converting measurements between these packages.

## 2. Materials and Methods

### 2.1. Study Population

We included 242 retinal photographs of eyes exempt of ocular condition from 133 patients from the Department of Ophthalmology of University Hospital of Grenoble Alps. All participants had a complete clinical examination, including refraction, visual acuity, measurement of intraocular pressure, slit-lamp examination for the anterior and posterior segment. All the funduscopy color photographs were 45-degree images and obtained with the same CR-2 Canon fundus camera (Canon™) after pupil dilation, at a resolution of 5184 × 3456 pixels. Overall, 19 images were excluded because of bad quality or because they were not centered on the optic disc. Good photographic quality was determined subjectively on the basis of the following criteria: focus (perceived sharpness of the vessel edge against the background retinal pigment epithelium), illumination (even and consistent lighting across the zone of vessel measurement), and color balance (equal saturation of color channels). Therefore, 223 images were considered for the analysis. This study complied with the Declaration of Helsinki guidelines for research involving human subjects. The Institutional review board (IRB# 5921) reviewed and approved the study protocol, as part of the ongoing prospective observational IMAGEYE cohort study. All study participants provided informed consent for all the ophthalmologic examinations and agreed that anonymized data could be used for clinical research.

### 2.2. Retinal Image Analysis

The measurement procedure has been described previously for VAMPIRE [22,23,24], IVAN [25], and SIVA [15]. Each image selected was analyzed with the three software tools. The interface of each software is briefly represented in Figure 1. A single trained software operator from the Grenoble team performed IVAN analysis and VAMPIRE analysis, while SIVA analysis was carried out independently by a trained operator from the Singapore Eye Research Institute. For IVAN, a standardized ARIC [25] grid was calibrated to a fixed size according to the photograph resolution and was then manually centered on the optic disc. For SIVA and VAMPIRE, the optic disc (and also the macula for VAMPIRE) was identified automatically by the software and adjusted manually in the case of incorrect location. Each package identified and traced the retinal arterioles and venules automatically. A trained grader then examined the traced vessels and manually corrected any incorrect vessel labels (artery or vein). IVAN and SIVA enable an operator to also modify the length of the traced vessels so as to obtain a better match.

In all cases, the vessels are analyzed in the Zone B annulus, from 0.5 to 1 disc diameter from the disc margin. The caliber of the six main arteries and veins was summarized with the CRAE and CRVE based on the revised Knudtson–Parr–Hubbard formula [26]. AVR was then calculated.

### 2.3. Pixel-to-Micron Conversion

We applied the commonly used pixel-to-micron conversion procedure based on the assumption that the adult human optic disc diameter is 1800 µm on average [27]. Using VAMPIRE, we computed the mean optic disc diameter in pixels of the entire set of images, while the IVAN and SIVA procedure is based on an image-converting factor (ICF) calculated in a subsample of images (10%).

### 2.4. Statistical Analysis

All analyses were performed with R (version 3.5.0) [28]. Results are presented as mean ± standard deviation (SD). Pearson’s correlation test was performed to evaluate the correlation between sets of measurements. The consistency in vessel diameter measurement between packages was estimated with intra-class correlation coefficients (ICC) using a two-way model, consistency definition, and single rater unit [29]. For Pearson’s correlation and ICC, the results were interpreted using the following scale: 0.00–0.39 = poor; 0.40–0.69 = moderate; and 0.70–1.00 = excellent. For ICC, single-measure coefficients and 95% confidence intervals (CIs) are reported.

To assess the agreement between software, we performed Bland–Altman analysis, where the 95% limits of agreement (LOA) were defined as mean difference ±1.96 × SD [30,31].

The three software applications were compared in pairs. A one-sample *t* test comparing the mean differences (between retinal vessel caliber of two different software applications) and zero value was run to indicate the presence of systematic bias. Pearson’s correlation analysis was conducted between the difference and the average (i.e., the axis of the Bland–Altman plot) to indicate the presence of proportional bias.

### 2.5. Conversion Algorithm

We propose a conversion algorithm (converting sets of measurements from different packages) derived using all the data of our sample and based on linear regression. For internal validation of our conversion model, we performed the bootstrapping procedure described by Labarère et al. [32]. Following this procedure, we generated a bootstrap sample of 1000 samples from our original sample. In order to assess the accuracy of our conversion algorithm, we studied the variation of the root-mean-square error (RMSE) in the bootstrap samples compared with the original RMSE in the original sample. Here, “error” means “difference between values from different packages.” The results are expressed in 95% CI, in which the RMSE of 95% of the samples is included.

## 3. Results

The 223 retinal photographs retained for analysis were obtained from 133 participants. The mean age of the participants was 47.9 ± 22.9 years and the sex ratio was 0.6 (50 men, 83 women). The mean axial length of the retinal photographs in the sample was 23.7 ± 0.9 mm. The characteristics of the participants are listed in Table 1.

Table 2 shows the mean absolute value and standard deviation of the CRAE, CRVE, and AVR measurements derived from VAMPIRE, IVAN, and SIVA for the 223 images. The agreement between values computed using VAMPIRE, IVAN, and SIVA is described by scatterplots and Bland–Altman plots (Figure 2, Figure 3 and Figure 4).

Statistical relationships are summarized in Table 3.

### 3.1. Between VAMPIRE and IVAN

The mean difference was 20.2 µm (95% LOA, −12.2–52.6 µm) for CRAE measurements, 21.0 µm (95% LOA, −17.5–59.5 µm) for CRVE measurements, and 0.023 (95% LOA, −0.12–0.16) for AVR values. Correlation was poor for CRAE and AVR (ICC = 0.29 and 0.35, respectively) and moderate for CRVE (ICC = 0.46). A proportional bias for CRAE and AVR was shown by a significant correlation between the difference and the average (*p* < 0.001 for CRAE and AVR), as represented in the Bland–Altman plot. This proportional bias was not observed for CRVE (*p* = 0.82).

### 3.2. Between VAMPIRE and SIVA

The mean difference was 36.6 µm (95% LOA, −12.8–60.4 µm) for CRAE measurements, 40.3 µm (95% LOA, 5.6–75.0 µm) for CRVE measurements, and 0.037 (95% LOA, −0.071–0.15) for AVR values. The correlation for CRAE and AVR between each software was poor (ICC = 0.37 and 0.38, respectively) and moderate for CRVE (ICC = 0.47). A significant proportional bias was observed for CRVE and AVR (*p* < 0.001) but not for CRAE (*p* = 0.09).

### 3.3. Between IVAN and SIVA

The mean difference was 16.4 µm (95% LOA, −4.25–37.0 µm) for CRAE measurements, 19.3 µm (95% LOA, 0.09–38.6 µm) for CRVE measurements, and 0.015 (95% LOA, −0.09–0.11) for AVR values. The correlation for CRAE and AVR between each software was good (ICC = 0.69 and 0.61, respectively) and excellent for CRVE (ICC = 0.83). A proportional bias was observed for all the retinal vessel parameters (*p* < 0.001 for all).

For each pair of packages and for each parameter, the presence of systematic bias was demonstrated by the significance (*p* < 0.001) of one-sample *t* tests comparing mean differences and zero value.

To determine a conversion method between measurements from different packages, we computed linear regression relationships for CRAE and CRVE measurements.

### 3.4. Between SIVA and IVAN

IVAN-derived SIVA CRAE = 0.4895 × IVAN-measured CRAE + 62.6017

IVAN-derived SIVA CRVE = 0.6518 × IVAN-measured CRVE + 56.8346

### 3.5. Between SIVA and VAMPIRE

VAMPIRE-derived SIVA CRAE = 0.3388 × VAMPIRE-measured CRAE + 79.0913

VAMPIRE-derived SIVA CRVE = 0.3503 × VAMPIRE-measured CRVE + 115.4105

Table 4 describes the comparisons between SIVA parameters and VAMPIRE-derived SIVA parameters or IVAN-derived parameters. In all cases, the mean of the differences between each sample was insignificant. Between SIVA measurements and VAMPIRE-derived SIVA approximates, the RMSE was 7.27 µm (6.48–8.15) for CRAE and 9.69 µm (8.71–10.79) for CRVE. Between SIVA measurements and IVAN-derived SIVA approximates, the RMSE was 4.88 µm (4.35–5.60) for CRAE and 5.19 µm (4.66–6.03) for CRVE.

## 4. Discussion

Our study analyzed the agreement and correlation between CRAE, CRVE, and AVR as measured by three widely used software tools—IVAN, SIVA, and VAMPIRE—with 233 fundus camera images of 133 retinas of healthy adults. Our analysis showed an excellent agreement between SIVA and IVAN, but a poor-to-moderate agreement between VAMPIRE and the two other software tools. In addition, we proposed a simple method for converting the measurements from one tool into estimates from another.

The evaluation of software tools such as SIVA, IVAN, and VAMPIRE can follow two different approaches. First, one can concentrate on morphological measurements of the retinal vascular tree, typically vessel diameters (as carried out here), tortuosity, junction/bifurcations, and fractal dimension. Second, one can study the association between vascular parameters and systematic parameters, or disease state or risk. In previous studies, the correlation with systematic parameters was found not to be significantly different between SIVA and IVAN [21], and between SIVA and VAMPIRE for the main vascular parameters (hypertension, systolic blood pressure, diastolic blood pressure, and HbA1c) [20,33].

For ophthalmologists, retinal vessel diameter can be considered a standard measurement from fundus images, such as, intraocular pressure, anterior chamber depth, corneal thickness, and axial length in routine practice. In this sense, using morphological measurements of retinal vessel diameters (the first approach) is of major interest. As a quantitative measure, this parameter provides objective information on the vascular retinal tree and could prove useful for the monitoring of ocular diseases, such as retinal vascular occlusion, diabetic retinopathy, or glaucoma. From this point of view, it may become mandatory to compare these retinal measurements in patients for vascular evaluation at baseline, during follow-up, and before and after treatment.

The agreement between IVAN and SIVA was higher than the agreement between IVAN and VAMPIRE or SIVA and VAMPIRE. When considering IVAN and SIVA, the mean difference in CRAE measurements of 16.4 µm (95% LOA, −4.25–37.0 µm) in our study was higher than −6.7 µm (95% LOA, −23.8–10.4 µm) reported in a previous study [21]; and the mean difference in the CRVE measurement of 19.3 µm (95% LOA, 0.09–38.6 µm) in our study was similar to that reported previously, −18.2 µm (95% LOA, −36.7–0.4 µm). Proportional bias was observed in both of these studies and, overall, both studies are consistent.

The poor agreement between SIVA and VAMPIRE has already been reported [20]. Interestingly, when comparing VAMPIRE and other software packages, the correlation was found to be higher for CRVE than for CRAE. This could be explained by the easier recognition of the venule edges by each software, due to their larger diameter and their more contrasted appearance in a retinal photograph. In our recent study comparing retinal photographs and adaptive optics (AO) imaging, we demonstrated a better agreement and correlation of vein measurements using IVAN when compared with the gold standard AO [34].

To counter the consequences of the discrepancy between measurements taken by different packages, we have proposed a conversion algorithm based on linear regression. The good correlation between SIVA and IVAN values of CRAE, CRVE, and AVR make this choice sensible, but less so if VAMPIRE values are involved. We did not use split-sampling for internal validation as reported in other publications [21] but we performed the bootstrapping procedure described by Labarère et al. [32]. The robustness of the algorithms is well assessed through the bootstrapping procedure. Between SIVA and IVAN, our algorithms are not strictly comparable to the algorithms previously reported (IVAN-derived SIVA CRAE = 0.7176 × IVAN-measured CRAE + 34.3984 IVAN-derived SIVA CRVE = 0.7102 × IVAN-measured CRVE + 44.8717) [21]. This can be explained by many factors, such as the retinal photograph resolution and compression, which could differ in the samples studied. The influence of these two parameters on IVAN measurements has been studied previously and found to be considerable [35].

The aforementioned findings highlight the need for standardization of retinal vascular imaging with fundus cameras (resolution, compression, size, instruments, acquisition protocol, quality among others) if we are to expect similar conclusions from a statistical analysis of measurements obtained with different software packages. One should consider conclusions of this study when analyzing images at high resolution (5184 × 3456 pixels) in a JPEG format. Standardization in the field of acquisition of fundus images and treatment of images by cameras is an urgent goal [36]. Development of deep-learning system [37] may produce fully automated measurement of retinal-vessel calibers and therefore may become a useful tool in the future for clinicians.

We should acknowledge some limitations of this study. First, the pixels-to-micron conversion procedure was the same for SIVA and IVAN, but not exactly the same for VAMPIRE, which could be a cause of systematic bias. Second, images were obtained from the instrument in a JPEG compressed format, which is known to distort image-based measurements; our results should be confirmed with uncompressed images [38,39,40]. Third, neither agreement nor correlation among packages attests to the accuracy of measurements with respect to objective ground truth. This kind of study has been reported elsewhere for the three software packages considered here. Fourth, SIVA and VAMPIRE generate a large number of parameters and we have studied only CRAE, CRVE, and AVR. Since several other parameters (e.g., tortuosity, fractal dimension) as well as width measurements in regions other than Zone B are used in biomarker studies, a more extensive comparison would be necessary for a proper assessment of the concordance between the three packages. Fifth, our sample is modest in size and comprises adults with healthy retinas. Larger studies including statistical adjustment modeling potentially important factors (e.g., camera operator, image quality, patient characteristics etc.) are required. Finally, the limited correlation of VAMPIRE values with those of SIVA and IVAN limits the validity of the conversion model, which could, however, have value in trend-oriented investigations (i.e., decreasing or increasing outcome).

## 5. Conclusions

In conclusion, our study on the agreement and correlation of measurements from the three most widely used software tools for retinal biomarkers found an excellent agreement between SIVA and IVAN and a poor-to-moderate agreement between VAMPIRE and the two other software tools.

Comparing measurements of the retinal vasculature obtained from different software tools remains a challenge requiring, arguably, a considerable standardization effort on algorithms, image acquisition protocols and quality at least. Standardization would guarantee not only consistent measurements but also, and importantly, comparable findings in terms of biomarkers for specific conditions obtained from statistical analysis. In our study, a solution suggested by our findings is to convert IVAN and VAMPIRE measurements into SIVA approximate equivalents before use for pooled data analysis.

## Figures and Tables

**Figure 1 diagnostics-12-01317-f001:**
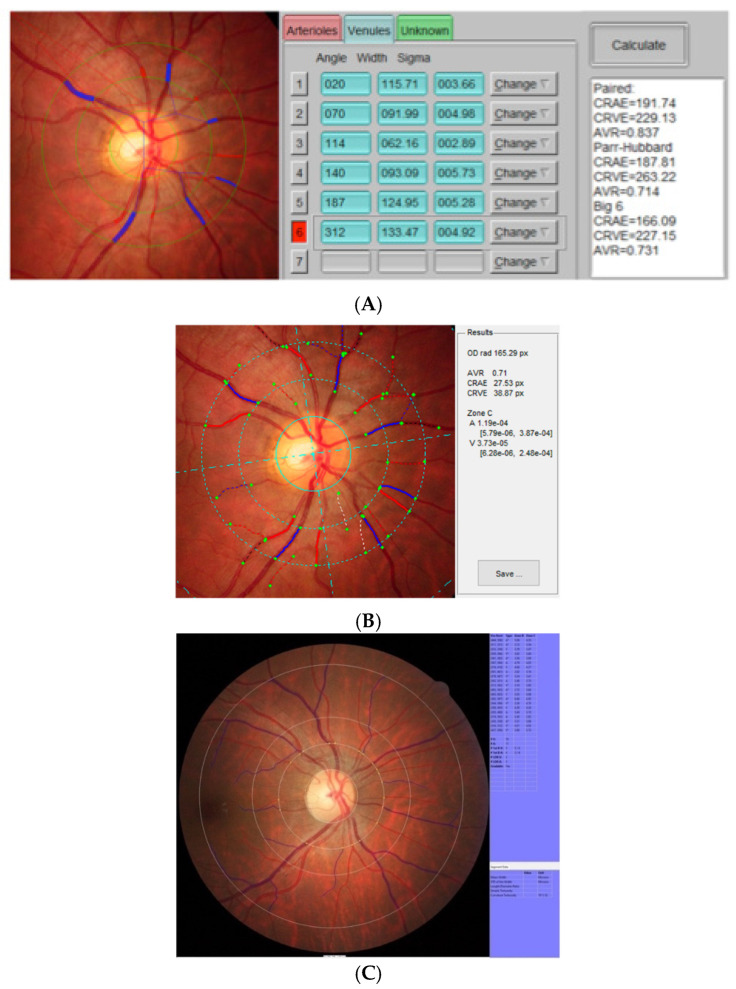
Presentation of the interface of each software. (**A**) VAMPIRE software interface, (**B**) IVAN interface, and (**C**) SIVA interface.

**Figure 2 diagnostics-12-01317-f002:**
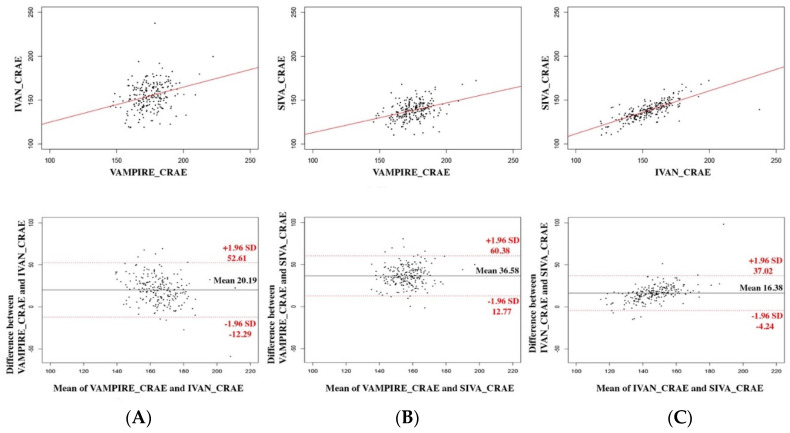
Agreement between each software for CRAE measurements. The three software applications are compared two by two: (**A**) IVAN and VAMPIRE, (**B**) SIVA and VAMPIRE, (**C**) SIVA and IVAN. Scatterplots are represented above with regression lines; Bland–Altman plots are represented below.

**Figure 3 diagnostics-12-01317-f003:**
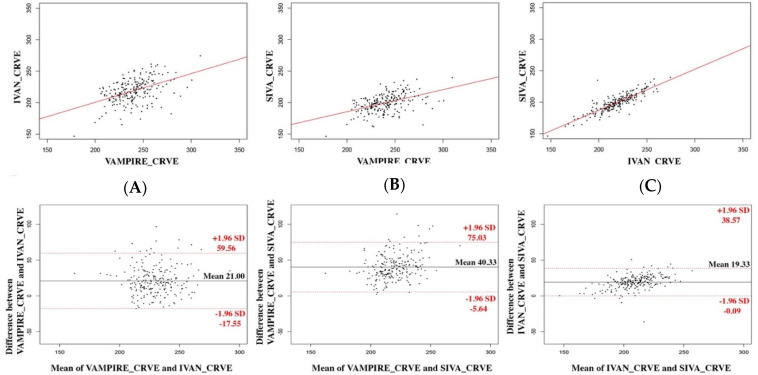
Agreement between each software for CRVE measurements. The three software applications are compared two by two: (**A**) IVAN and VAMPIRE, (**B**) SIVA and VAMPIRE, (**C**) SIVA and IVAN. Scatterplots are represented above with regression lines; Bland–Altman plots are represented below.

**Figure 4 diagnostics-12-01317-f004:**
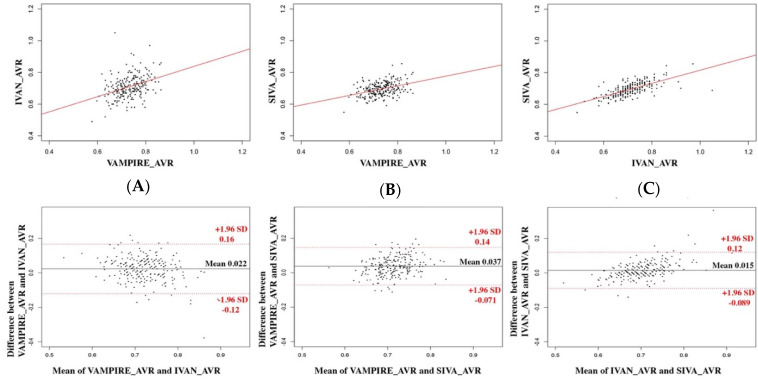
Agreement between each software for AVR. The three software applications are compared two by two: (**A**) IVAN and VAMPIRE, (**B**) SIVA and VAMPIRE, (**C**) SIVA and IVAN. Scatterplots are represented above with regression lines; Bland–Altman plots are represented below.

**Table 1 diagnostics-12-01317-t001:** Clinical characteristics of the study population.

Characteristics	Participants (n = 133) Mean (SD)/n (%)
Age, years	47.9 (22.9)
Sex ratio male/female	0.6
Male	50 (37.6)
Female	83 (62.4)
Axial length, mm	23.7 (0.9)
Intra-ocular pressure, mmHg	14.3 (2.8)
Systolic blood pressure, mmHg	125.9 (12.5)
Diastolic blood pressure, mmHg	75.7 (9.7)
Heart rate, bpm	73.1 (10.7)
Body mass index, kg/m^2^	22.5 (7.1)

Bpm: beat per minute.

**Table 2 diagnostics-12-01317-t002:** Retinal vessel parameters with VAMPIRE, IVAN, and SIVA. CRAE: central retinal artery equivalent, CRVE: central retinal vein equivalent, AVR: arteriole-to-venule ratio.

Retinal Vessel Parameter		Mean (±SD)	
	VAMPIRE	IVAN	SIVA
CRAE (µm)	174.9 (±11.3)	154.7 (±15.7)	138.3 (±10.2)
CRVE (µm)	239.7 (±19.1)	218.7 (±18.8)	199.4 (±14.2)
AVR	0.73 (±0.054)	0.71 (±0.072)	0.69 (0.04)

**Table 3 diagnostics-12-01317-t003:** Agreement between software applications according to retinal vessel parameter. * *p* value of one-sample *t* tests comparing mean difference and zero (systematic bias). ^†^
*p* value of Pearson’s correlation coefficients of the regression line (proportional bias). *r*, Pearson’s correlation between the parameter values obtained with the two software applications considered. AVR: arteriole-to-venule ratio, LOA: limits of agreement.

VAMPIRE and IVAN	Mean Difference	95%LOA	*p* Value *	*r*	ICC	IC 95	*p* Value ^†^
CRAE (µm)	20.2	(−12.2 to 52.6)	<0.001	0.29	0.27	(0.15–0.39)	<0.001
CRVE (µm)	21.0	(−17.5 to 59.5)	<0.001	0.46	0.46	(0.35–0.56)	0.82
AVR	0.023	(−0.12 to 0.16)	<0.001	0.36	0.35	(0.23–0.46)	<0.001
VAMPIRE and SIVA							
CRAE (µm)	36.6	(12.8 to 60.4)	<0.001	0.38	0.37	(0.25–0.48)	0.09
CRVE (µm)	40.3	(5.6 to 75.0)	<0.001	0.47	0.45	(0.33–0.55)	<0.001
AVR	0.037	(−0.071 to 0.15)	<0.001	0.38	0.37	(0.25–0.48)	<0.001
IVAN and SIVA							
CRAE (µm)	16.4	(−4.25 to 37.0)	<0.001	0.75	0.69	(0.61–0.75)	<0.001
CRVE (µm)	19.3	(0.09 to 38.6)	<0.001	0.86	0.83	(0.78–0.87)	<0.001
AVR	0.015	(−0.09 to 0.11)	<0.001	0.68	0.61	(0.52–0.68)	<0.001

**Table 4 diagnostics-12-01317-t004:** Assessment of the robustness of the conversion algorithm using bootstrapping. The root-mean-square error between (RMSE) SIVA measurements and VAMPIRE-derived SIVA approximates or IVAN-derived SIVA approximates (RMSE in µm) is included in the ICC interval obtained in 1000 bootstrap samples. AVR: arteriole-to-venule ratio, CRAE: central retinal artery equivalent, CRVE: central retinal vein equivalent.

**Between SIVA Measurements and VAMPIRE-Derived SIVA Approximate**			
	RMSE (µm)	95% CI	
CRAE	7.27	(6.48–8.15)	
CRVE	9.69	(8.71–10.79)	
**Between SIVA Measurements and IVAN-Derived SIVA Approximate**			
	RMSE (µm)	95% CI	
CRAE	4.88	(4.35–5.60)	
CRVE	5.19	(4.66–6.03)	

## Data Availability

Data may be available on request.

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
