# Peer review of "Concordance between SIVA, IVAN, and VAMPIRE Software Tools for Semi-Automated Analysis of Retinal Vessel Caliber"

_diagnostics, 2022, doi:10.3390/diagnostics12061317_

Round 1

Reviewer 1 Report

Interesting research. However, there are some possible improvements:

- 24/36 references are from the authors of the propose paper, which is not appropriate.
- Flaws are stated in the manuscript. However, nothing is proposed to improve solutions or to guide further research. See lines 275-279. The problems are actually the most interesting part and should be addressed further.

Author Response

- 24/36 references are from the authors of the propose paper, which is not appropriate.

We added the following references of studies using IVAN software, written by other teams. We acknowledge that most publications using VAMPIRE and SIVA softwares originated from collaborations between different teams, including those which developed these softwares, i.e. coauthors of this paper.

Page 2, line 48:

  • Iwase A, Sekine A, Suehiro J, et al. A New Method of Magnification Correction for Accurately Measuring Retinal Vessel Calibers From Fundus Photographs. Invest Ophthalmol Vis Sci 2017;58:1858–64. doi:1167/iovs.16-21202
  •  
  • Wei F-F, Zhang Z-Y, Petit T, et al. Retinal microvascular diameter, a hypertension-related trait, in ECG-gated vs. non-gated images analyzed by IVAN and SIVA. Hypertension Research 2016;39:886–92. doi:1038/hr.2016.81
  • Dervenis N, Coleman AL, Harris A, et al. Factors Associated With Retinal Vessel Diameters in an Elderly Population: the Thessaloniki Eye Study. Invest Ophthalmol Vis Sci 2019;60:2208–17. doi:1167/iovs.18-26276

- Flaws are stated in the manuscript. However, nothing is proposed to improve solutions or to guide further research. See lines 275-279. The problems are actually the most interesting part and should be addressed further.

Reviewer 2 Report

I think the authors submitted an interesting manuscript with a decent amount of experimental results. My comments are the followings.

1.) Figure 1 is not ideal. Probably a better solution would be to split Figure 1 into 3 separate figures to illustrate better the software interfaces. In Figure 1, I was not able to observe the details of the interfaces.

2.) The authors have written that 242 retinal photographs from 133 patients were collected. The authors could illustrate the manuscript with some retinal photographs. What should a medical doctor observe on these retinal photographs? I think it would be great for readers with engineering or computer science background.

3.) Do the authors have experience about the effects of image quality? In the literature, the quality assessment of natural and retinal images is a very hot research topic? Papers for citation: No-reference image quality assessment with convolutional neural networks and decision fusion, Evaluation of retinal image quality assessment in different color-spaces. Can the examined software packages handle images with low quality? Does a software package outperform the other in this respect?

4.) In software comparison, it is very important to analyze the requirements of the project. However, requirement analysis is rather unclear in this manuscript. What are the main requirements for such software packages? The authors should break down the system into functional components and analyze each element and the system as a whole. 

Author Response

I think the authors submitted an interesting manuscript with a decent amount of experimental results. My comments are the followings.

1.) Figure 1 is not ideal. Probably a better solution would be to split Figure 1 into 3 separate figures to illustrate better the software interfaces. In Figure 1, I was not able to observe the details of the interfaces.

Figures have been uploaded with a higher definition, as requested.

2.) The authors have written that 242 retinal photographs from 133 patients were collected. The authors could illustrate the manuscript with some retinal photographs. What should a medical doctor observe on these retinal photographs? I think it would be great for readers with engineering or computer science background.

Illustrations of measurements using different softwares are already in figure 1.  We do not think that additional figures will be useful for the readers, especially as there is no retinal disease in these healthy subjects.

3.) Do the authors have experience about the effects of image quality? In the literature, the quality assessment of natural and retinal images is a very hot research topic? Papers for citation: No-reference image quality assessment with convolutional neural networks and decision fusion, Evaluation of retinal image quality assessment in different color-spaces. Can the examined software packages handle images with low quality? Does a software package outperform the other in this respect?

As reported in the manuscript, we included healthy patients without corneal disease, cataract in order to obtain high quality fundus images (page 2, line 74).

The main quality issue in the image set used in our study is compression. We address this in Section 4, Discussion, last paragraph, second limitation mentioned. We have added the following comment and a reference:

We have reported elsewhere a quantitative analysis of the effect of JPEG compression on the values of retinal parameters computed by VAMPIRE (M R K Mookiah, S Hogg, T J MacGillivray, E Trucco on behalf of the INSPIRED consortium: On the quantitative effects of compression of retinal fundus images on morphometric vascular measurements in VAMPIRE. Computer Methods and Programs in Biomedicine,  vol 202, 105969, 2021.)

4.) In software comparison, it is very important to analyze the requirements of the project. However, requirement analysis is rather unclear in this manuscript. What are the main requirements for such software packages? The authors should break down the system into functional components and analyze each element and the system as a whole. 

The comparison in the study is not meant as a comparison between software packages from an architectural of module-by-module point of view. Instead, as it is typical of papers on biomarkers, the comparison is focussed on the values of parameters computed by the various packages. The parameters have been defined clearly elsewhere, so that the comparison is consistent. The parameters considered in our study summarize the width of the 6 largest veins or arteries in Zone B, defined clearly in the paper ( last paragraph before Figure 1). Vessel  width, in turn, is defined as the width of the blood column and the algorithms estimating it in each of the three systems have been reported elsewhere, including in papers cited in our manuscript.

Round 2

Reviewer 1 Report

You haven't addressed "However, nothing is proposed to improve solutions or to guide further research."

Image quality is not well addressed.  What are definitions and ways of calculation for image quality measures?

Author Response

You haven't addressed "However, nothing is proposed to improve solutions or to guide further research.

The main action needed to counteract the variability of retinal vascular measurements, in our opinion, is a standardization effort. We note that the revised manuscript already suggested this important line of further research (lines 389-397).

We have also amended the conclusions to highlight our recommendations (lines 426-432).

Image quality is not well addressed.  What are definitions and ways of calculation for image quality measures?

Photographic quality criteria has been added lines 81-85.

Reviewer 2 Report

The authors improved the manuscript which contains interesting results. I think it is suitable for publication.

Author Response

(The authors gave the same response as above.)
